# Machine Learning Approach to Metabolomic Data Predicts Type 2 Diabetes Mellitus Incidence

**DOI:** 10.3390/ijms25105331

**Published:** 2024-05-14

**Authors:** Andreas Leiherer, Axel Muendlein, Sylvia Mink, Arthur Mader, Christoph H. Saely, Andreas Festa, Peter Fraunberger, Heinz Drexel

**Affiliations:** 1Vorarlberg Institute for Vascular Investigation and Treatment (VIVIT), A-6800 Feldkirch, Austria; axel.muendlein@vivit.at (A.M.); arthur.mader@lkhf.at (A.M.); christoph.saely@vivit.at (C.H.S.); anderasfesta@icloud.com (A.F.); heinz.drexel@vivit.at (H.D.); 2Central Medical Laboratories, A-6800 Feldkirch, Austria; smink@mzl.at (S.M.); pfraunberger@mzl.at (P.F.); 3Faculty of Medical Sciences, Private University of the Principality of Liechtenstein, FL-9495 Triesen, Liechtenstein; 4Department of Internal Medicine III, Academic Teaching Hospital Feldkirch, A-6800 Feldkirch, Austria; 5Vorarlberger Landeskrankenhausbetriebsgesellschaft, Academic Teaching Hospital Feldkirch, A-6800 Feldkirch, Austria; 6Drexel University College of Medicine, Philadelphia, PA 19129, USA

**Keywords:** ML, machine learning, artificial intelligence, diabetes, incidence, metabolomics, support vector machine, accuracy

## Abstract

Metabolomics, with its wealth of data, offers a valuable avenue for enhancing predictions and decision-making in diabetes. This observational study aimed to leverage machine learning (ML) algorithms to predict the 4-year risk of developing type 2 diabetes mellitus (T2DM) using targeted quantitative metabolomics data. A cohort of 279 cardiovascular risk patients who underwent coronary angiography and who were initially free of T2DM according to American Diabetes Association (ADA) criteria was analyzed at baseline, including anthropometric data and targeted metabolomics, using liquid chromatography (LC)–mass spectroscopy (MS) and flow injection analysis (FIA)–MS, respectively. All patients were followed for four years. During this time, 11.5% of the patients developed T2DM. After data preprocessing, 362 variables were used for ML, employing the Caret package in R. The dataset was divided into training and test sets (75:25 ratio) and we used an oversampling approach to address the classifier imbalance of T2DM incidence. After an additional recursive feature elimination step, identifying a set of 77 variables that were the most valuable for model generation, a Support Vector Machine (SVM) model with a linear kernel demonstrated the most promising predictive capabilities, exhibiting an F1 score of 50%, a specificity of 93%, and balanced and unbalanced accuracies of 72% and 88%, respectively. The top-ranked features were bile acids, ceramides, amino acids, and hexoses, whereas anthropometric features such as age, sex, waist circumference, or body mass index had no contribution. In conclusion, ML analysis of metabolomics data is a promising tool for identifying individuals at risk of developing T2DM and opens avenues for personalized and early intervention strategies.

## 1. Introduction

Type 2 diabetes mellitus (T2DM) poses a significant and escalating global public health challenge, with its incidence continually increasing. Addressing this growing epidemic requires innovative strategies for early detection and intervention to prevent long-term complications such as cardiovascular diseases, nephropathy, neuropathy, and retinopathy.

Metabolomics, the detailed analysis of small-molecule metabolites within biological systems, stands out as a potent tool for uncovering metabolic changes and disease phenotypes. By mapping the entire metabolite spectrum in biological samples, it offers a detailed view of physiological and pathological states, shedding light on the biochemical pathways implicated in diabetes [1,2,3]. The metabolome’s complexity and its sensitivity to genetic and environmental factors make it an ideal candidate for exploring the mechanisms of disease onset and progression.

Artificial Intelligence (AI) significantly boosts risk prediction capabilities and has shown efficacy in detecting diabetes [4,5] and its complications [6]. Machine learning (ML), with its proficiency in deciphering large, complex datasets, represents a promising approach to leveraging metabolomics in T2DM research [7]. ML algorithms excel at identifying subtle patterns and correlations within metabolomic data that traditional statistical methods might miss. Such capabilities herald new possibilities in forecasting T2DM incidence, facilitating the identification of individuals at risk through their metabolomic profiles. This synergy between ML models and metabolomic data has introduced an innovative method for understanding the metabolic disruptions preceding T2DM [8], potentially enabling earlier diagnosis and tailored intervention strategies.

The associations between T2DM and cardiovascular disease (CVD) are well established, and CVD is the leading cause of mortality and morbidity in T2DM patients [9]. This prospective study thus aims to predict the risk of new-onset T2DM in a cohort of cardiovascular risk patients, utilizing metabolomic data and ML. By melding advanced ML techniques with comprehensive metabolomics datasets, we intend to identify metabolic markers indicative of T2DM incidence. Our efforts will not only enhance our comprehension of T2DM’s pathophysiological underpinnings but also highlight the transformative potential of integrating metabolomics with ML in advancing early detection and prevention strategies. Ultimately, this research may help to initiate a new chapter in diabetes care, characterized by the synergy of predictive analytics and personalized medicine, to effectively counter the T2DM epidemic.

## 2. Results

### 2.1. Patient Characteristics

This study evaluated 279 non-diabetic participants, as detailed in Table 1. The median age of the cohort was 65 years, with an interquartile range (IQR) of 59–73 years. Throughout the 4-year follow-up period, 11% of these participants (n = 32) developed T2DM. A comparative analysis revealed that, aside from glucose levels, there were no significant differences in characteristics between those who developed T2DM and those who did not. Regarding HbA1c, both groups were in a prediabetic state [10], though HbA1c itself was not significantly different between both groups.

### 2.2. Feature Selection

In the preliminary stage of our analysis, we employed recursive feature elimination (RFE) to identify both crucial and less significant variables. Table 2 presents a summary of the outcomes from various RFE models, including Random Forest, TreeBag, Naïve Bayes, and the Caret default. Based on the results from the Random Forest-based RFE model, we opted for a narrowed selection of variables (n = 77) and, as an alternative based on the results from both the TreeBag and Caret default models, also for the whole set of variables (n = 362) for further in-depth analysis. Regarding the plots depicted in Figure 1, there was no distinct cutoff, thus confirming our approach to proceed with both the selectively reduced and the full sets of variables. Notably, across both selections, ceramides and hexoses (six-carbon sugars) emerged as the most important features according to the top five rankings, highlighting their potential significance in predicting T2DM incidence.

### 2.3. ML Model Comparison

Finally, ML was applied using a set of ten models as specified in Section 4. Table 3 outlines the performance metrics for the evaluated models. As the number of positive outcomes was limited in our study (n = 32), sensitivity and accuracy may be misleading and were not ideal performance measures. We thus prioritized “F1-score” and “balanced accuracy” as the most critical metrics due to the outcome imbalance observed in our dataset. We tested all models with both the selectively reduced set of variables (n = 77) and the full set (n = 362).

Using the smaller set of variables, the “svmLinear2” model—a Support Vector Machine with linear kernels—emerged as the top performer, yielding the highest F1 score (50%). This model also demonstrated notable balanced accuracy (72%) and precision (50%), effectively identifying 50% of patients who developed T2DM (sensitivity) and 93% of those who did not (specificity), leading to an (unbalanced) accuracy of 88%. The Multi-Layer Perceptron (“mlp”) model followed, marking the second-best performance with an F1 score of 40% and balanced accuracy of 66%.

Utilizing the full set of variables (n = 362), models’ performance varied, with no consistent trend of improvement or decline observed when compared to the reduced set of variables (n = 77). Notably, the Rotation Forest model stood out, ranking third overall in performance with an F1 score of 40% and balanced accuracy of 63%. A comparison of ROC curves for all models is presented in Appendix A.

### 2.4. Important Features Ranking

The twenty most important features of the top-ranked “svmLinear2” model according to the “VarImp()” function are depicted in Figure 2. The top features were hexoses, amino acids (glycine, isoleucine, and the amino acid derivative kynurenine), and bile acids (chenodeoxycholic acid, CDCA and deoxycholic acid, DCA). A comparison of these features regarding their amount in patients who developed T2DM (positive outcome) and those who did not (negative outcome) illustrates that indeed nearly all of them were significantly different between both patient outcomes (Appendix A) and correlate with this outcome (Appendix A). However, some features, including certain ceramides and glycerophospholipids, demonstrated high collinearity (Appendix A). Furthermore, taking into account all top 20 ranked features of each model (generated by “VarImp()”), we have calculated a feature importance score (FIS) based on in-model ranking and frequency (Table 4). Hexoses were again the top-ranked feature (FIS = 354). However, when referring only to single metabolites, the amino acid glycine was top-ranked, followed by another amino acid, isoleucine; the bile acids CDCA, glucoursodeoycholic acid (GUDCA), and DCA; and a ceramide (N-C15:1-Cer, containing a pentadecenoic acid). When referring to metabolite classes instead, bile acids had the highest count (FIS = 1012), followed by ceramides (FIS = 928), amino acids (FIS = 766), and hexoses (FIS = 354).

Moreover, the Shapley Additive Explanation (SHAP) method offered an alternative, model-agnostic view of variable significance (Figure 3). Comparing results from SHAP and the “VarImp()” function, there were some similarities regarding the inclusion of hexoses and ceramides in the top-ranked features, but also some discrepancies, particularly a high ranking of lactic acid and several acylcarnitines (tiglylcarnitine, decenoylcarnitine, hexadecenoylcarnitine, tetradecadienylcarnitine, and carnitine).

Notably, the anthropometric variables age, sex, waist circumference, waist–hip ratio, and body mass index (BMI) were almost completely ignored by the applied ML algorithms and did not show up either in the SHAP or in the “VarImp()”-generated list of top-ranked features (Figure 2 and Figure 3), suggesting a limited impact.

### 2.5. Validation Model Output with a Linear Regression Approach

For comparison, we applied a more traditional statistical modeling approach known for its linear nature and interpretability: Lasso-Regression. This method utilized the same preselected set of variables generated by RFE (n = 77). As a result, Lasso Regression excluded 66 variables and used 11 variables for prediction (Appendix A). Overall, 10 out of these 11 variables were also part of the top 20 most important variables identified by the VarImp() function (Table 3) or the SHAP method (Figure 3). Of note, no ceramides or glycerophospholipids (both featuring high collinearity) were part of the Lasso-Regression model. Given this difference compared to the more complex ML models, the predicting performance of the Lasso-Regression approach was worse (F1 score = 19%, balanced accuracy = 45%).

## 3. Discussion

### 3.1. Main Findings

In our investigation, ten distinct ML algorithms were employed to evaluate a comprehensive metabolomic dataset, augmented with anthropometric data, for the prediction of T2DM incidence among cardiovascular risk patients undergoing coronary angiography over a four-year period. Among the models analyzed, the “svmLinear2” model—a Support Vector Machine with a linear kernel—stood out, delivering the highest F1 score (50%) and balanced accuracy (82%). This model demonstrated a 50% sensitivity rate, accurately identifying half of the patients who developed T2DM. Furthermore, it achieved a precision of 50%, indicating that half of the T2DM onset predictions were correct. Remarkably, the model successfully classified 93% of the individuals who did not develop T2DM during the follow-up.

The most important variables out of these metabolites were bile acids, ceramides, amino acids, and hexoses. Contrarily, anthropometric measures—despite being part of the analysis—did not significantly contribute to the predictive accuracy of the model.

### 3.2. The Role of Metabolites in Predicting New-Onset Diabetes

Remarkably, ML was capable of predicting T2DM incidence within four years, although patients who did develop T2DM and those who did not were not strikingly different in terms of their anthropometric or HbA1c data. A recent study highlighted the challenge in classifying HbA1c levels regarding the risk of developing diabetes, noting that many individuals are already classified as high-risk under current HbA1c thresholds [12]. They suggested that raising the threshold to an HbA1c value of 6.0% could improve the positive predictive value (=precision) to 12.4%. In our study, the difference in HbA1c levels between patients who developed T2DM during the follow-up period and those who did not was minimal and not statistically significant (5.85% vs. 5.70%, *p* = 0.052), remaining in any case below the suggested thresholds. Unlike the low predictive performance observed with HbA1c alone, our findings demonstrate that for a clinically relevant patient group, which is already at high cardiovascular risk undergoing coronary angiography, a more comprehensive risk prediction is possible when incorporating a broader dataset, particularly metabolomics data.

Machine Learning, in contrast, has identified a set of metabolites, including bile acids, ceramides, amino acids, and hexoses, which were indeed significantly different in patients who develop T2DM. Noteworthy, these metabolites were identified by the complex ML models, and in part by the more simple Lasso Regression. Bile acids, as one of these, play a pivotal role in lipid metabolism. Recent studies have uncovered that they act as signaling molecules, influencing glucose metabolism and insulin sensitivity, thus having an impact on the development of diabetes [8,13,14]. Similarly, ceramides—molecules composed of a fatty acid linked to a sphingosine backbone—have emerged as potent predictors of risk [15,16]. Initially investigated for their association with cardiovascular disease [17], recent studies have illuminated their significant influence on energy metabolism and diabetes [18,19]. However, we noticed that, in contrast to complex ML models which handle collinearity and imply feature interactions, ceramides as well as glycerophospholipids were excluded in the Lasso-Regression model, given their high collinearity. In a previous prospective study involving 2776 patients, Liu et al. [3] demonstrated, using targeted metabolomics and Lasso Regression, that certain metabolic features—including isoleucine, tyrosine, and lactate, which were also among our top 20 features—significantly outperformed glucose in predicting T2DM in the majority of patients. A recent ML-based study identified kynurenine (also among our top 20 features) as one of the most important predictors of T2DM incidence in Chinese patients [8].

Our ML study has now demonstrated for the first time that, alongside hexoses, these metabolite classes can predict the onset of diabetes in non-diabetic cardiovascular risk patients.

That said, it is worth mentioning that a previous ML analysis by Shojaee-Mend et al. has identified age, waist–hip ratio, and BMI as the most important features predicting diabetes [20]. BMI, especially those values >40, contributes to a high risk of diabetes [21]. The BMI in our study was much lower, with an IQR of 25–31. Thus, neither BMI nor any further anthropometric measures emerged as top features in our model predictions. This divergence suggests that when metabolomic data are included, these anthropometric features, given that they are not extreme values, may play a less significant role, underscoring the potential superior predictive power of metabolomic markers in forecasting diabetes onset. Conversely, it has recently been demonstrated that molecular markers, including 19 metabolites, improved the prognostic performance of ML models beyond that of classical risk factors [22]. This accentuates the findings of our study; metabolomic features could offer a more nuanced understanding of diabetes progression. It is important to note, however, that while certain metabolomic features are identified or corroborated here as top predictors, interpreting these features and understanding their biological significance may require further investigation. As a result, validating these features and understanding their interplay could mark a significant step forward in diabetes prevention and management strategies.

### 3.3. The Role of ML Models in Predicting New-Onset Diabetes

In the present study, we reached the highest prediction performance when applying a Support Vector Machine algorithm to a dataset that had been scaled down, applying a recursive elimination of the dataset’s variables (in our setting from 362 to 77). Compared to Lasso Regression, which represents a more traditional linear regression approach, and is robust and easier to interpret but excludes features with high collinearity, machine learning models, particularly Support Vector Machines, effectively manage collinearity and capture feature interactions. Maybe this is the reason for the much higher predictive performance demonstrated in our study. In a recent study revisiting the Pima Indian diabetes dataset, the Support Vector Machine algorithms had also been identified to have the highest accuracy in identifying existing diabetes, though other algorithms, including Multi-Layer Perceptron, Random Forest, decision trees, and Naïve Bayes, also showed accuracies around 70% [23]. A different study applying the same Pima Indian diabetes dataset has generated an improved artificial Neural Networks model attaining the highest accuracy [24].

These examples underline the complexity of comparing ML models, especially with diverse datasets (age, race, disease, etc.). Apart from that, we must mention that, other than the high values for balanced accuracy in our study, the F1 score is not very high (50.0% for the top model), but it is still in range with a comparable study reporting F1 scores of 44.9–47.1% [20]. However, this [20] and the majority of other studies have reported the prediction of existing diabetes. Our focus, in contrast, was on predicting the onset of newly developed diabetes in non-diabetic individuals over four years, adding another layer of challenge distinct from identifying existing diabetes. Nonetheless, our findings suggest the SVM algorithm’s robustness in diabetes prediction, particularly within datasets rich in metabolomics data.

Interestingly, our RFE process revealed that reduced feature sets, in particular the one identified through Random Forest, optimized SVM performance. Yet, not all models showed improved performance with fewer features, echoing previous research indicating Random Forest’s limitations in handling omics data with many correlated variables [25]. This suggests that while SVM stands out in our study, future investigations might uncover algorithms better suited to varying datasets. Moreover, we anticipate that expanding the dataset—both in patient numbers and balancing class ratios—could significantly enhance prediction accuracy and generalizability.

### 3.4. Strengths and Limitations

This study has strengths and limitations. A particular strength of this study is its metabolomics dataset which initially comprised over 500 metabolites. An additional strength is the very well-characterized cohort of cardiovascular disease patients, who were all enrolled under identical prerequisites in the same tertiary care center. Moreover, we analyzed prospective data regarding the incidence of diabetes. Since the diagnosis of existing diabetes is routinely conducted using glucose marker testing, predicting the risk of new-onset diabetes in originally non-diabetic patients with a high cardiovascular risk is more ambitious and of high clinical relevance.

One limitation of our study is the fact that we selected only Caucasian patients with an elevated cardiovascular risk undergoing coronary angiography. Therefore, the results are, of course, not representative in view of the general population nor necessarily applicable to other patients or other ethnicities, which might impact the model performance. In addition, though we validated the training set-based prediction on test set data, which was randomly generated from the whole dataset, validation in a different cohort ideally characterized by comparable metabolomics would be commendable. Further, we did not apply HbA1c or the formal prediabetes status for predicting T2DM incidence, as both variables are closely linked or defined by glucose values, which are already part of the metabolomic profile in terms of hexoses. Furthermore, apart from the metabolomic differences between non-diabetic patients who developed T2DM and those who did not, as investigated in the present study, we suppose that patients who develop T2DM during a shorter follow-up may exhibit an even more meaningful metabolomic profile than those who develop T2DM later. Unfortunately, the incidence rate in our cohort of cardiovascular risk patients is too limited to conduct such an analysis. Finally, we have data on the prescription of drugs but not on adherence to the respective medical treatment, which also may impact the outcome.

## 4. Materials and Methods

### 4.1. Study Subjects and Patient Selection for Metabolomic Analysis

Patients were selected from a coronary angiographically characterized Caucasian cohort, recruited between 2005 and 2009, as previously detailed [26]. They were consecutively enrolled for angiography at our tertiary care hospital (Academic Teaching Hospital Feldkirch) to evaluate established or suspected coronary artery disease (CAD), excluding those with acute coronary syndromes or Type 1 diabetes. Basic clinical measurements and laboratory analyses were conducted at the Central Medical Laboratories Feldkirch, thoroughly described elsewhere [27]. Venous blood samples, taken after a 12 h fast, underwent immediate basic laboratory testing. Serum samples were then aliquoted and frozen at −80 °C to facilitate metabolomic analysis, safeguarding against repeated freeze–thaw cycles.

Diagnosis of T2DM followed American Diabetes Association (ADA) guidelines [10] requiring one or more of the following: fasting plasma glucose ≥126 mg/dL (7.0 mmol/L), plasma glucose ≥200 mg/dL (11.1 mmol/L) post-oral glucose tolerance test (OGTT), hemoglobin A1c (HbA1c) ≥ 6.5% (47.5 mmol/mol), or previously confirmed diabetes. Diabetic status was evaluated at recruitment and again at a four-year follow-up to determine incidence rates.

For the metabolomics assay, we randomly selected 407 baseline patient samples from the above-described total study population, provided they had completed the four-year follow-up visit. We applied a targeted quantitative metabolomic approach to analyze the stored serum samples (BIOCRATES Life Sciences AG, Innsbruck, Austria) using liquid chromatography (LC)–mass spectroscopy (MS) and flow injection analysis (FIA)–MS as described previously [28]. In total, 535 compounds were analyzed. Alongside these metabolomics features, we also included anthropometric features age, sex, BMI, waist circumference, and waist-to-hip ratio in the dataset. Patients with known T2DM or who were diagnosed with T2DM at baseline were excluded (n = 128) from further analysis. Therefore, metabolomics data for 279 non-diabetic patients were available. The selection of patients as well as the data preprocessing and analysis scheme is depicted in Appendix A.

### 4.2. Data Analysis Steps

#### 4.2.1. Preprocessing and Recursive Feature Elimination

Data preprocessing was performed using R version 4.2.2 and the Caret package [11]. Initially, we removed 179 metabolites from the dataset due to missing values exceeding 30%. The dataset was then randomly divided (75:25) into training (209 samples) and testing (70 samples) groups using the “createDataPartition()” function. The training set was utilized for model development, while the testing set helped evaluate the model’s performance in predicting T2DM incidence. Missing values in the training set were addressed using the “knnImpute” method from the “preprocess()” function. Categorical variables underwent one-hot encoding with “dummyVars()”, and numerical variables were normalized via the range method of the same function. Prior to feeding the preprocessed data to the ML algorithms, we assessed variable importance through recursive feature elimination (RFE).

RFE, a wrapper method, iteratively builds models to evaluate and discard the least important features. It thereby allows for the assessment of feature importance in the context of the model and the data and identifies features (variables included in the metabolomic dataset) that are more relevant than others to predict the target variable (T2DM incidence), resulting in the enhanced predictive accuracy of the final model. Utilizing the “rfeControl()” and “rfe()” functions, we explored four distinct algorithm sets within the RFE framework—Random Forest (rfFuncs), Tree Bagging (treebagFuncs), Naive Bayes (nbFuncs), and Caret Default (caretFuncs)—to ensure a comprehensive and reliable feature selection. Each set underwent a 10-fold cross-validation repeated 5 times via the “repeatedcv” method, enhancing the robustness of feature selection. The outcomes of RFE, illustrating the pivotal features, are depicted in Table 2 and Figure 1.

Following RFE, the testing dataset was processed using the same imputation and normalization techniques as the training set to maintain consistency across data preparation stages.

#### 4.2.2. Statistical Analysis

Differences in baseline characteristics were tested for statistical significance with the Chi-squared tests for categorical and Jonckheere–Terpstra tests for continuous variables. Correlation analyses were performed by calculating non-parametric Spearman rank correlation coefficients. These analyses were performed with SPSS 28.0.0.0. for Windows (SPSS, Inc., Chicago, IL, USA). Multicollinearity analysis was performed by calculating the Variance Inflation Factor using the “car” package in R [29]. Lasso (Least Absolute Shrinkage and Selection Operator) Regression was performed using the “glmnet” package [30].

#### 4.2.3. Managing Data Imbalance

The initial dataset exhibited a significant imbalance in outcomes, with only 32 subjects developing diabetes compared to 247 who did not. Recognizing that such an imbalance could adversely affect the performance of many machine learning algorithms [31], we addressed this issue by increasing the representation of the minority class. This was achieved through the application of the “up” sampling method, integrated within the “trainControl()” function of the caret package in R. This approach balanced the dataset by augmenting the number of records for subjects who newly developed diabetes, thereby enhancing the potential for more accurate and equitable model performance.

#### 4.2.4. ML Analysis for Predictive Classification Modeling

For the binary classification of “T2DM developed” vs. “T2DM not developed”, we utilized either the smaller set of features selected through RFE or the full set of features. Model generation employed the “train()” function from the Caret package, incorporating a range of machine learning algorithms identified for their high performance in recent studies [32,33,34,35,36,37,38,39,40,41] and through a meticulous pre-screening process. We tested ten distinct models: Random Forest (“rf”) [32], an implementation of the Rotation Forest algorithm with complexity parameter tuning (“rotationForestCP”) [33], a Naive Bayes classifier (“naïve_bayes”) [34], a Multi-Layer Perceptron (“mlp”) [35], a feedforward Neural Network with a Principal Component Analysis step (“nnet”) [36,37], a Bootstrap Bagging of decision trees (“treebag”) [38], two variations of Support Vector Machines with linear kernels (“svmLinear” and “svmLinear2”) [39], an Extreme Gradient Boosting method, tailored for tree-based models (“xgbtree”) [40], and a further open-source Gradient Boosting on decision trees method (“catboost”) [41,42]. A detailed overview is given in Table 3.

To enhance model accuracy and minimize fitting errors, we leveraged the “trainControl()” function for meticulous training protocol definition, including cross-validation strategies, evaluation metrics, and data preprocessing methods. Specifically, we adopted a 5-fold cross-validation strategy, activated class probability estimation for binary classification, and implemented up-sampling for the minority class. While default parameters were maintained for general model training aspects, hyperparameter optimization was streamlined through the Caret package’s capability to automatically adjust hyperparameters via a predefined grid for each algorithm. Post-optimization, the “expand.grid()” function allowed for the fine-tuning of model parameters on the enriched training dataset, leading to a rigorous evaluation of model performance.

#### 4.2.5. ML Model Performance Evaluation

The evaluation metrics for the various models drew from true positive (TP), false positive (FP), true negative (TN), and false negative (FN) values. These metrics encompassed sensitivity (also known as recall; calculated as TP/(TP + FN)), specificity (TN/(TN + FP)), precision (or positive predictive value; TP/(TP + FP)), accuracy (the sum of TP and TN divided by the total number of cases; (TP + TN)/(TP + TN + FP + FN)), balanced accuracy (the average of sensitivity and specificity; (sensitivity + specificity)/2), and the F1 score (the harmonic mean of precision and sensitivity; 2 × (precision × sensitivity)/(precision + sensitivity)). These metrics were generated utilizing the “confusionMatrix()” function in the Caret package.

To assess the significance of various variables, we employed the “varImp()” function within the Caret package. This analysis enabled us to rank variables by their impact, offering insights into the overall variable importance within each model.

Another method for interpreting ML model outputs is the SHAP approach [42]. For this, we utilized both the kernelshap and shapviz packages in R [43]. The “kernelshap()” function, a model-agnostic method for computing SHAP values, employs weighted linear regression to estimate each feature’s contribution to individual predictions. SHAP values visualization was achieved through the “shapviz()” function, enhancing the understanding of the model decision-making processes.

## 5. Conclusions

In conclusion, metabolomic data are superior to anthropometric data, and applying ML for their analysis is a promising new tool for identifying individuals at risk of developing T2DM as it opens avenues for personalized and early intervention strategies.

## Figures and Tables

**Figure 1 ijms-25-05331-f001:**
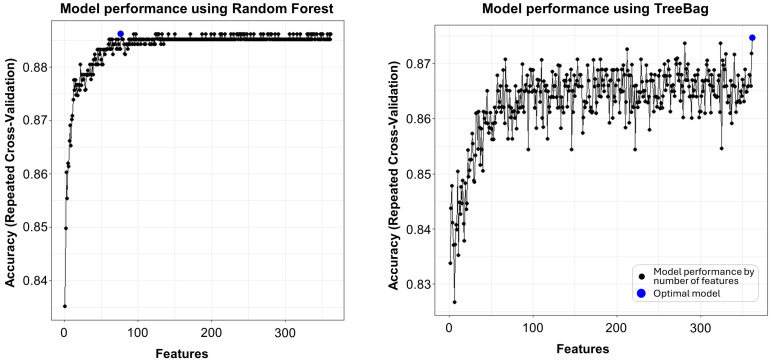
Identifying important variables by recursive feature elimination. Recursive feature elimination helps to identify important and less important variables and to define the optimal size of ML models, as summarized in Table 2. The plot represents the output of an RFE process generating different models (black dots). It depicts the relation between the different feature subset sizes (=number of available variables (1–362)) for modelling and the resulting performance metric (accuracy = (true positive + true negative)/(true positive + false positive + true negative + false negative)). Using Random Forest (**left**) and TreeBag algorithms (**right**) as functions in RFE, the best models (highlighted as blue dots) were calculated to have 77 and 362 variables, respectively. The process involves repeated cross-validation (method = repeatedcv (10-fold repeated 5 times)) to evaluate the performance of feature subsets. The plots were generated by ggplot using the Caret package in R (CRAN, R [11]).

**Figure 2 ijms-25-05331-f002:**
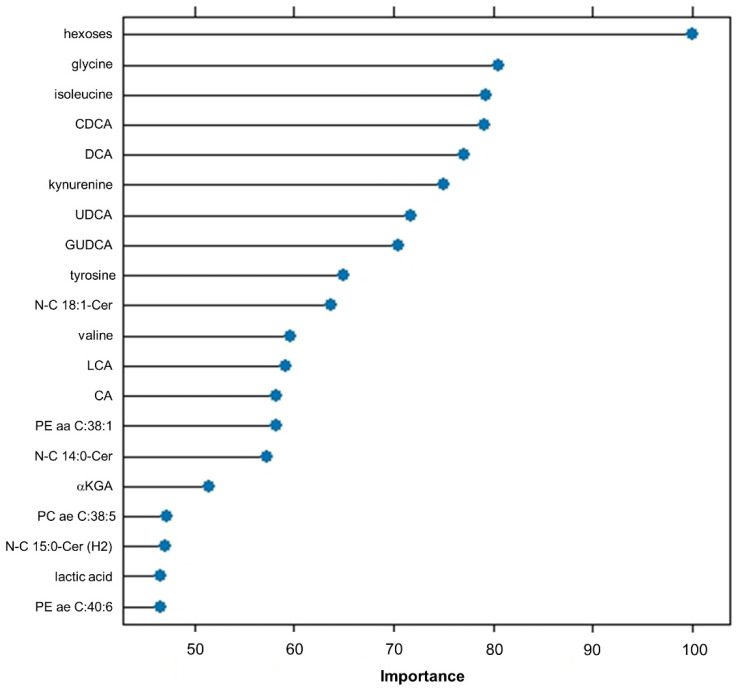
Importance of model variables. The figure depicts the most important variables and the respective importance scores according to the “VarImp()” function in Caret (CRAN, R [11]). Here, the top 20 variables of the “svmLinear2” model are displayed, including hexoses, amino acids (glycine, isoleucine, tyrosine, valine), bile acids (chenodeoxycholic acid = CDCA, deoxycholic acid = DCA, ursodeoxycholic acid = UDCA, glycoursodeoxycholic acid = GUDCA, litocholic acid = LCA, cholic acid = CA), ceramides (N-C 18:1-Cer, N-C 14:0-Cer, N-C 15:0-Cer(H2)), energy metabolism intermediates (alpha-ketoglutaric acid, lactic acid), glycerophospholipids (PE aa C38:1, PC ae C38:5, PE ae C40:6), and a biogenic amine (kynurenine).

**Figure 3 ijms-25-05331-f003:**
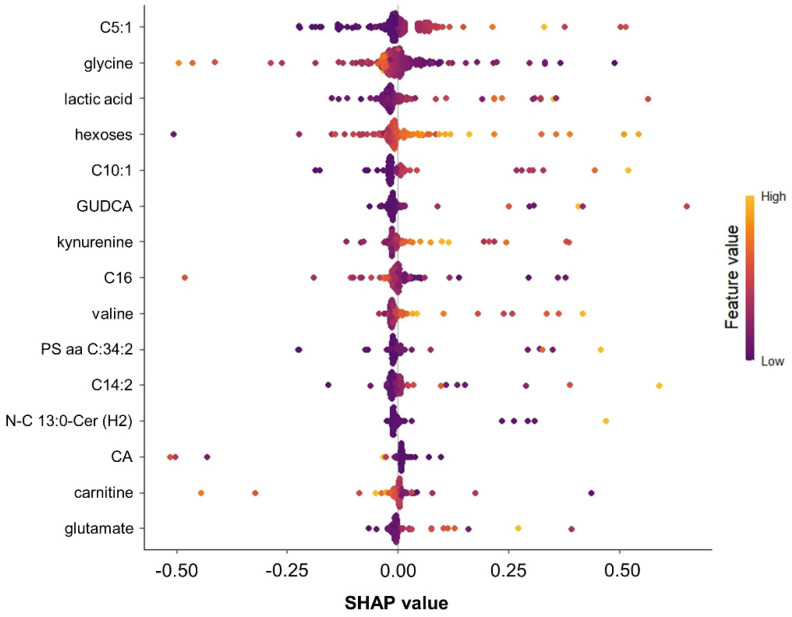
SHAP diagram of feature importance. The beeswarm plot illustrates the most important features (variables) and the contribution of these individual features to the model’s output using Shapley Additive Explanation (SHAP) values. Each dot represents a SHAP value for a feature and a specific data point, indicating the magnitude and direction of the feature’s impact on the model’s prediction relative to the baseline. The *y*-axis demonstrates the variable name, in order of importance from top to bottom, and the *x*-axis the SHAP value scale. It indicates how large the impact of the respective variables is on the model output (T2DM incidence). The gradient color indicates the original value for that variable. C5:1 represents tiglylcarnitine, C10:1 decenoylcarnitine, C16 hexadecanoylcarntine, C14:2 tetradecadienylcarnitine, GUDCA glycoursodeoxycholic acid, CA carnitine, PS aa C:34:2 a phosphatidylserine with a diacyl bond, and N-C13:0-Cer(2H) a dihydroceramide.

**Table 1 ijms-25-05331-t001:** Baseline patient characteristics.

	Total	No T2DM Incidence	T2DM Incidence	*p*-Value
	n = 279	n = 247	n = 32	
Age (years) *	65 [59–73]	65 [59–74]	68 [66–72]	0.460
Male sex (%) *	50	49	59	0.269
BMI (kg/m^2^) *	28 [25–31]	28 [25–30]	29 [27–31]	0.155
Waist circumference (cm) *	99 [93–106]	99 [92–106]	102 [96–109]	0.083
Waist–hip ratio *	0.97 [0.91–1.00]	0.96 [0.91–1.00]	0.98 [0.94–1.02]	0.203
LDL-C (mg/dL)	134 [107–162]	134 [105–162]	135 [115–165]	0.495
Fasting glucose (mg/dL)	98 [90–106]	97 [90–105]	108 [99–115]	<0.001
HbA1c (%)	5.70 [5.50–5.90]	5.70 [5.50–5.90]	5.85 [5.70–6.00]	0.052
Hypertension (%)	89	87	97	0.115
Smoking, current (%)	19	17	28	0.143
Statin treatment (%)	42	41	44	0.791

Dichotomous data are given as proportion, and continuous data (all not normally distributed) are given as median and interquartile range [IQR]. Differences between patients who developed T2DM during the four-year follow-up and patients who did not develop T2DM during follow-up were tested with Chi-squared tests for categorical and Jonckheere–Terpstra test for continuous variables. Anthropometric variables included in ML model generation are highlighted by asterisks.

**Table 2 ijms-25-05331-t002:** Feature selection.

Function (ML Algorithm)	Validation Technique	Range of Input Variable Subsets (n)	Size of Best Model (n)	Top 5 Variables
rfFuncs (Random Forest)	Repeated CV (10-fold repeated 5 times)	1 to 362	77	N-C15:0-OH-Cer, N-C19:0-Cer, N-C15:1-Cer, Hexoses, C14:2
treebagFuncs (Treebag)	Repeated CV (10-fold repeated 5 times)	1 to 362	362	Hexoses, TLCAS, N-C15:1-Cer, Gly, C14:2
nbFuncs (Naive Bayes)	Repeated CV (10-fold repeated 5 times)	1 to 362	1	Hexoses
caretFuncs (Caret default)	Repeated CV (10-fold repeated 5 times)	1 to 362	362	Hexoses, Gly, TLCAS, N-C15:1-Cer, C10:1

Recursive feature elimination (RFE) was used for feature selection in subsets ranging from 1 to 362 variables. Four different algorithms were used and validated by repeated cross-validation (CV). N-C19:0-Cer, N-C15:1-Cer, and N-C15:0(OH)-C represent ceramides and hydroxyacyl-ceramides, respectively. C14:2 and C10:1 represent the acylcarnitines tetradecadienylcarnitine and decenoylcarnitine. Gly represents glycine and TLCAS the sulfated bile acid taurolithocholic acid sulfate. Hexoses represent monosaccharides with six carbon atoms.

**Table 3 ijms-25-05331-t003:** Comparison between models and metrics.

MLalgorithm (Method)	Variable Set (n)	F1	Accuracy	Balanced Accuracy	Precision	Sens.	Spec.	AUC
Random Forest (“rf”)	362	0%	88.4%	50.0%	0%	0%	100%	0.5789
Random Forest (“rf”)	77	22.2%	89.9%	56.3%	100%	12.5%	100%	0.5840
Rotation Forest with complexity parameter tuning (“rotationForestCp“)	362	40.0%	91.3%	62.5%	100%	25.0%	100%	0.7039
Rotation Forest with complexity parameter tuning (“rotationForestCp“)	77	16.7%	85.5%	53.8%	25.0%	12.5%	95.1%	0.7295
Naive Bayes classifier (“naïve_bayes”)	362	20.8%	11.6%	50.0%	11.6%	100%	0%	0.5000
Naive Bayes classifier (“naïve_bayes”)	77	0%	85.5%	48.4%	0%	0%	96.7%	0.4344
Multi-Layer Perceptron (“mlp“)	362	38.5%	76.8%	70.6%	27.8%	62.5%	78.7%	0.6516
Multi-Layer Perceptron (“mlp“)	77	40.0%	87.0%	65.5%	42.9%	37.5%	93.4%	0.7213
Feedforward Neural Networks with a Principal Component Step (“nnet”)	362	33.3%	82.6%	63.0%	30.0%	37.5%	88.5%	0.6814
Feedforward Neural Networks with a Principal Component Step (“nnet”)	77	30.8%	73.9%	63.5%	22.2%	50.0%	77.1%	0.6680
Bootstrap bagging of decision trees (“treebag”)	362	16.7%	85.5%	53.8%	25.0%	12.5%	95.1%	0.5666
Bootstrap bagging of decision trees (“treebag”)	77	23.5	81.2%	56.8%	22.2%	25.0%	88.5%	0.625
Support Vector Machine with linear kernels (“svmLinear“)	362	34.8%	78.3%	66.0%	26.7%	50.0%	82.0%	0.5430
Support Vector Machine with linear kernels (“svmLinear“)	77	30.8%	73.9%	63.5%	22.2%	50.0%	77.0%	0.6352
Support Vector Machine with 2 linear kernels (“svmLinear2“)	362	34.8%	78.3%	66.0%	26.7%	50.0%	82.0%	0.5389
Support Vector Machine with 2 linear kernels (“svmLinear2“)	77	50.0%	88.4%	71.7%	50.0%	50.0%	93.4%	0.7254
Tree-based extreme gradient boosting (“xgbTree“)	362	36.4%	89.9%	61.7%	66.7%	25.0%	98.4%	0.6680
Tree-based extreme gradient boosting (“xgbTree“)	77	34.8%	78.3%	66.0%	26.7%	50.0%	82.0%	0.6270
Gradient boosting on decision trees (“catboost”)	362	0%	88.4%	50.0%	0%	0	100.%	0.4795
Gradient boosting on decision trees (“catboost”)	77	33.3%	87.0%	60.9%	50.0%	25.0%	96.7%	0.6311

The result table gives an overview of the applied models and their respective performance metrics. The F1 score (2 × (precision × sensitivity)/(precision + sensitivity) was selected as the most important metric.

**Table 4 ijms-25-05331-t004:** Overall ranking of model features.

Rank	Feature	Metabolite Class	FIS
1	Hexoses	Hexoses	354
2	Gly	Amino acids	309
3	Ile	Amino acids	259
4	CDCA	Bile acids	258
5	GUDCA	Bile acids	213
6	DCA	Bile acids	211
7	N-C15:1-Cer	Ceramides	177
8	UDCA	Bile acids	140
9	Kynurenine	Biogenic amines	137
10	N-C14:1-Cer (2H)	Ceramides	105
11	N-C19:0(OH)-Cer (2H)	Ceramides	102
12	Tyr	Amino acids	96
13	N-C18:1-Cer	Ceramides	95
14	Val	Amino acids	82
15	N-C14:0-Cer	Ceramides	77
16	GCDCA	Bile acids	76
17	C16	Acylcarnitines	70
18	PE aa C34:0	Glycerophospholipids	56
19	N-C25:1-Cer	Ceramides	55
20	N-C12:0-Cer	Ceramides	45

The table lists the most important features according to the feature importance score (FIS), which sums up the ranking score of the top 20 features (=20/rank) in all models described in Table 3.

## Data Availability

The data that support the findings of this study are available from the corresponding author upon reasonable request.

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
