# Peer review of "Machine Learning Approach to Metabolomic Data Predicts Type 2 Diabetes Mellitus Incidence"

_ijms, 2024, doi:10.3390/ijms25105331_

Round 1

Reviewer 1 Report

Comments and Suggestions for Authors

The authors conducted a study in which machine learning algorithms were applied to predict the 4-year risk of developing type 2 diabetes mellitus (T4DM). Their analysis included ML prediction and feature importance analysis. Additional experimentation is required due to the model's low accuracy. The novelty of this paper lies in the use of previously unexplored data. However, the data appear to have been adopted from earlier studies. Thus, the novelty of this study may limit its publication.

The following are detailed comments:

1. I disagree with the success of their machine learning model due to low F1 scores and other indicators.

2. The paper is overly technical. Even if they include function names, readers may not understand them.

3. Make the code and data available to readers to ensure reproducibility.  

4. Authors should discuss their findings using recent publications, such as Journal of the American College of Cardiology 83, 2026-2026. 

Author Response

Dear Editor, dear Reviewers,

We sincerely appreciate your insightful comments and suggestions, which have greatly contributed to the improvement of our manuscript. Below, you will find our point-by-point response to your comments. We have addressed all the points raised and have thoroughly revised the manuscript to include additional data and analyses, such as correlation analysis and Lasso Regression. Consequently, we believe that the study's findings are now more comprehensible to the general readership. Notably, we have also revised the structure of the manuscript and removed subheadings from the abstract as requested by the editor.

In light of these revisions, we believe the manuscript is now suitable for publication. We are more than willing to make any further adjustments if deemed necessary.

Once again, thank you for your time and valuable insights. We look forward to the possibility of contributing to your esteemed journal

Sincerely,

Andreas Leiherer

Editor

We have recently sent you the reviewers comments for revision.

During this time, we also kindly ask you to revise the following couple of issues, in accordance with the journal formatting guidelines.

  1. Headings. Please note that the heading order in IJMS Communcation type papers should be: 1. Introduction, 2. Results 3. Discussion, 4.

Materials and Methods. Taking this into consideration, we kindly ask you to move the"Materials and Methods" section after "Discussion".

Thanks for this comment. The editorial team already made this change.

  1. Abstract. During this time, we kindly ask you to also remove the headings ("Background", "Methods", "Results", "Conclusions") from the Abstract section, as per IJMS guidelines.

Thanks for this comment. We have removed these headings.

Reviewer#1

The authors conducted a study in which machine learning algorithms were applied to predict the 4-year risk of developing type 2 diabetes mellitus (T4DM). Their analysis included ML prediction and feature importance analysis. Additional experimentation is required due to the model's low accuracy. The novelty of this paper lies in the use of previously unexplored data. However, the data appear to have been adopted from earlier studies. Thus, the novelty of this study may limit its publication.

The following are detailed comments:

  1. I disagree with the success of their machine learning model due to low F1 scores and other indicators.

Thank you for your comment. It's important to contextualize the F1 scores reported. A previous study (Shojaee-Mend 2024, Health Inform Res 30(1), 73-82; already cited as reference 28 in the original version)  predicted diabetes prevalence—not incidence, as in our study—and reported F1 scores ranging from 44.9% to 47.1% for their best model. While our F1 score of 50.0% may not seem exceptionally high, it is an improvement over these figures. Additionally, our study achieved high values for balanced accuracy and specificity (as well as overall accuracy), which are critical in the context of incidence prediction where the balance between sensitivity and specificity is crucial to effectively identify new cases. Furthermore, our model addresses the unique challenges associated with predicting the incidence of diabetes, which involves forecasting new cases over time. This task generally presents fewer cases but greater complexity compared to predicting prevalence, which could partly explain the modest F1 scores. These points have been elaborated in the revised manuscript.

  1. The paper is overly technical. Even if they include function names, readers may not understand them.

We thank the reviewer for this comment. We organized the paper structure and technical description according to recent publications and we feel that this is (i) state-of-the-art and (ii) necessary to follow and understand our findings. On the other hand, of course, we want to improve the understandability of our manuscript. In line with the comments of reviewer#2 who suggested adding a simpler method (Lasso-Regression) and a comprehensible characterization of the identified important variables to make sure that all readers understand the findings, we have thoroughly revised our manuscript:

  • Comparison of predictors between patients with positive and negative outcomes (lines 143-147, supplementary table 1)
  • Simple correlation analysis of identified predictors and outcome (lines 143-147, supplementary table 2)
  • Linear Regression (Lasso) approach, which is easier to understand and to interpret (lines 193-201, lines 233-236, lines 273-276, supplementary table 3)

  1. Make the code and data available to readers to ensure reproducibility.  

The code will be made available in an online repository or supplement according to the journal requirements.  

  1. Authors should discuss their findings using recent publications, such as Journal of the American College of Cardiology 83, 2026-2026. 

Regarding your reference suggestion,  American College of Cardiology, Volume 83 page 2026, I found that these recent publications are abstracts of the recent ACC24 meeting in Atlanta. I attended this meeting for all three days and I was completely focused on machine learning to get my manuscript state-of-the-art. However, using the acc24 app and looking for abstracts I have not found the combination “machine learning” AND “diabetes”, “machine learning” AND “diabetes”, or “prediction” AND “diabetes”, though there were many other ML abstracts.

Nonetheless, we carefully revised the manuscript and added further relevant studies, including a metabolomics study (reference suggested by reviewer#2) and a recent study on the prediction of T2DM incidence in Chinese patients.   

Reviewer#2

The study conducts a thorough analysis using machine learning algorithms to predict the 4-year risk of developing type 2 diabetes mellitus (T2DM) by leveraging targeted quantitative metabolomics data. Initially comprising 279 cardiovascular risk patients without T2DM, the cohort underwent comprehensive baseline assessments, including anthropometric data and targeted metabolomics. Over the follow-up period, 11.5% of patients developed T2DM. Among the 362 variables measured, 77 features were selected using recursive feature elimination. The SVM model emerged as the top performer among conventional machine learning algorithms, exhibiting superior performance based on F1-score. However, there are major concerns regarding the comprehensiveness and novelty of this work, as outlined below.

Major comments:

  1. This study didn’t seem to stand out. This paper is not the first one to use machine learning and metabolomics to predict the risk of T2DM, there are lot of similar papers that use metabolomics to predict T2DM. It’s not clear to me what’s the novelty of this paper. Authors need to highlight their unique contribution to the field.

Thanks for this valuable comment. The reviewer is right that there are already many papers using metabolomics to predict T2DM. However, just a small amount has used ML for prediction. The unique feature of our paper is, however, that we predict the incidence of new diabetes in a prospective study. To the best of our knowledge, we have not found such a study in the current literature predicting the risk apart from one single study in Chinese patients (ref 8 in the revised manuscript).

We have carefully revised our manuscript now. In the new version, we explicitly mention the novelty of our study more clearly, in particular the prospective character identifying new-onset T2DM in CVD patients. This is now part of the aim section in the introduction (lines 59-62). Together with the strengths and limitations paragraph, particularly describing the strength of our study setting (lines 302-309), we now hope that the readers are aware of the novelty of our data.

In line 59, authors write “This study aims to evaluate ML's capacity to utilize metabolomic data for predicting T2DM onset.” If that’s the case, authors should consider conducting a meta-analysis that incorporates as many samples as possible. The sample characteristics available in this study seem inadequate for achieving this aim. For example, the study is based on a specific cohort of cardiovascular risk patients at very old age (median age 65), which may limit the generalizability of the findings to other populations, so perhaps the highlight should be the prediction of T2DM risk specifically among cardiovascular risk patients.

This point needs to be addressed significantly to clarify the novelty and unique contribution of the study to the existing literature on predicting T2DM using metabolomics and machine learning.

First of all, we are very grateful for this great comment which helped to specify our study aim and significantly improved the manuscript. Of course, you are absolutely right, this sentence in line 59 of the original version was pretty misleading and we have deleted it in the revised manuscript. According to your comment, we now state in the revised aim section of the introduction that this is a “ prospective study [which] aims to predict the risk of new-onset T2DM in a cohort of cardiovascular risk patients…” (lines 59-62). (We would like to point out that 65 years is not very old for cardiovascular risk patients, it is a classical “real-world situation” age in clinics).

  1. To me, it seems this study has a small sample size problem. In other metabolomics impact on diabetes studies like Lu et al. (2016) and Liu et al. (2017), the number of patients exceeds several thousand. Here, the total sample size is 279, with only 32 of them developing T2DM. After the training/testing split, only an average of 6.4 diabetes patients will appear in the testing set, which makes sensitivity and specificity estimation very inaccurate. It would greatly enhance the study if the authors could include more patients. Additionally, while the inclusion criteria specify only Caucasian patients with elevated cardiovascular risk undergoing coronary angiography, it would be beneficial to include patients from other racial backgrounds if available. This could allow for additional analyses to determine whether race impacts model performance.

Thank you for your insightful comment and the references provided. Unfortunately, I was unable to locate the specific study by Lu et al. (2016) that fits the criteria. I found one study by Lu et al. in "Diabetology" (2016) with only 197 patients and another in "Diabetes Care" addressing gestational diabetes in 817 pregnant women. A 2017 study by Liu et al. in "Metabolomics" did indeed recruit 2,776 patients and measured 261 markers using Lasso-Regression, which we discuss in the revised manuscript (lines 241-244). However, there is just one study using ML and metabolomics to predict the incidence of T2DM (Su et al, J Adv Res, 2023), and this study has comparable characteristics (578 metabolites / 392 patients).

Regarding the inclusion of diverse racial backgrounds, our study region, Vorarlberg, predominantly consists of Caucasian residents, making it challenging to recruit a racially varied sample. This limitation is acknowledged in the revised "strengths and limitations" section (lines 310-313).

Moreover, the characterization of each sample with targeted metabolomics as we did (customized panel including several kits including 535 metabolites) as provided by the company Biocrates (Innsbruck, Austria) costs up to 400€ (426$) per sample. Therefore, the total costs for our patients were 163,000€ and for the non-diabetic 279 patients of our study at least 112,000€. This was a huge investment for a non-profit research institute. It is completely impossible for us to include more patients and test them in the same way.

We also acknowledge the potential inaccuracies in sensitivity and specificity estimation due to the small number of diabetes cases in our study. Instead of solely relying on accuracy, which can be unreliable with few outcome events, we tested the null hypothesis that the model's accuracy is no better than the “no information rate” using the p-value. We excluded models when we did not have sufficient evidence to conclude that the model performs better than trivially guessing the most frequent class. (We did not mention these statistical prerequisites to avoid making the paper overly technical, but we mention the limitation of accuracy early in the results section (line 117-119)). Therefore, we used other measures: Balanced accuracy and the F1 score. The former measure uses the average of sensitivity and specificity, providing an overall effectiveness of the model across both classes. As we had high values our model is good at detecting both positives and negatives. The F1 score, which is the harmonic mean of precision and recall, is a further measure preventing inaccuracy with a stronger focus on positives. Though this score is not very high, it is sufficient and even a bit higher than in a comparable study (ref 18). We mention this in the revised version of the manuscript (lines 284-288).

  1. Authors need to conduct further analysis on features. Regardless of whether they use Variable Importance (VarImp) or SHAP methods, which may not be easily understood by all readers, simpler methods should be employed to provide more intuitive information on feature importance. For instance, authors should perform association tests on each of the features individually, similar to what they did for baseline patient characteristics in Table 1. Another method to consider is lasso regression. The key point here is to provide an assessment of feature importance that has a clear meaning that most people can comprehend. Furthermore, feature correlations should be calculated. At line 53, “ML algorithms excel at identifying subtle patterns and correlations within metabolomic data that traditional statistical 53 methods might miss”. However, there’s no such analysis performed given the author point out this important issue. Given that the number of features is larger than the number of samples, dimension reduction techniques are needed.

Thanks for these excellent comments and suggestions. As suggested by the reviewer, we generated a new table similar to the patient characteristics in table 1.  This table contains all features identified by the VarImp() function in figure 2. The readership is now able to easily see that identified features/metabolites are indeed different between positive and negative patient outcomes. Moreover, we also added a further table in which correlation was calculated between the highly important features and the outcome. It is supplementary table 1 and 2 in the revised manuscript. I am not sure whether these two tables should be part of the main text, because too many tables and figures make it a bit difficult to read in the provided print design. I have pasted it in the supplementary part, but I am also fine when adding it to the main text. Maybe the editor could help here. Anyway, regarding the main text, the results and method sections were adapted accordingly (lines 384-389) and we now address these data in the revised version of the discussion section (lines141-145 and 230-232).

Furthermore, we also applied Lasso-Regression as suggested by the reviewer (This more traditional statistical method, which is based on linear regression and likely to be more comprehensible to the readership, was used for comparison with the more complex ML algorithms. Regarding the comparison with traditional statistics, we now mention that the ML model had a higher performance in predicting T2DM incidence than the traditional linear regression represented by  Lasso-Regression. Results are summarized in supplementary table 3. We added a new paragraph to the results section of the revised manuscript (lines 193-201), discussed it (lines 235-236, 243-248, and 273-276), and mentioned it in the methods section (lines 389 -390, ref 28).

Finally, we want to emphasize that the recursive feature elimination (rfe) we applied has already reduced the dataset, being far smaller than the number of patients (77 vs. 279). Apart from that, many ML models we tested use dimensional reduction techniques (e.g. principal component analysis, feature selection, feature extraction…) and also Lasso-Regression is characterized by its dimensionality reduction step excluding “less important” features

While certain metabolomic features are identified as top predictors, the interpretation of these features and their biological significance may require further investigation.

We thank the reviewer for highlighting this important point of the discussion. These concluding remarks are very concise and we have added it to the revised discussion section (lines  264-269)

  1. Supplementary information was not correctly uploaded. Authors need to provide complete supplementary information for further assessment.

We apologize for this and will contact the editorial office about what was going wrong. In the revised version of the manuscript, we added further data including characterization of important features, correlation analysis, and Lasso-Regression output.

Minor comments:

  1. Term “SHAP” was first mentioned in line 158, however complete definition was given at line 167 in figure caption and line 377. All abbreviations should be list full name as 1st time mentioned in main text.

We thank the reviewer for accurate reading and pointing out that mistake. Now, it is explained in full name upon first mentioned.

  1. In Figure 1, dots with different color should add a figure legend indicate what they represent.

In the original version, it is already explained that black dots represent different models and blue dots the best models Maybe you missed it in your version.

  1. In line 218, the authors conclude that anthropometric features like BMI play a less significant role compared to metabolomic data. However, I noticed that the range of BMI of patients included in this study has an interquartile range (IQR) from 25 to 31. According to Gray et al. (2015), when BMI is greater than or equal to 40, hazard ratios of T2DM show a significant increase. I doubt this conclusion is due to the lack of completeness of extreme values, thus the conclusion may not be comprehensive. Authors need to check their anthropometric distributions to see if their range is capable of detecting potential differences before making the conclusion that metabolomic data has superior power for T2DM prediction.

This is a very sophisticated and smart comment, thank you very much! We checked the range of BMI. It ranges from 17 to 41 /with only 2 patients >40). Thus, your statement is true. Our patient cohort does not sufficiently cover the severely obese ones. We thus have relativized our conclusion and now mention that this is only applicable to our patients and patients who are not obese (revised discussion, lines 253-261).  

  1. A narrowed selection of 77 variables is an important point, because it enables best model performance, it should be mentioned in abstract.

We thank the reviewer for this valuable comment. We now mention this finding in the abstract of the revised manuscript. (lines 25-26).

  1. Since it’s a 4-year follow up studies, it would be interesting to learn if years of onsite can be also predicted.

This is a very good question, one that I have often been asked during presentations of these data. Unfortunately, we checked only at follow-up visits and had only two of these. Given the limited number of outcomes, it did not make sense to further stratify.

  1. Relevant to previous points, since diagnostics were made up to 4 years, will there be patients potentially develop T2DM in future? Will short follow up time impact model performance? Is longer follow up years needed?

This is a very interesting point and a good question. In our cardiovascular risk patients, we see a relatively high prevalence of T2M patients (>30% of our total study population). Many of them are just diagnosed to have diabetes which they did not know previously. This is very relevant for risk prediction and medical treatment. The incidence of T2DM in those who have no T2DM at baseline (<70% of our total study population) is very important as well and thus was the aim of our study. We believe patients who develop T2DM during a shorter follow-up may have an even more meaningful metabolomic profile than those who developed T2DM later (in life). Unfortunately, the incidence rate is yet too small to make such a stratification. This is a very good point and we have added it to the limitations section of the revised manuscript (lines 319-324).   

  1. At line 215, “Machine learning has now demonstrated for the first time that, alongside hexoses, these metabolite classes can predict the onset of diabetes in non-diabetic patients.” Is this the first paper that demonstrate machine learning can predict onset of diabetes? If its not please don’t make such over-statement.

We meticulously reviewed the literature as described above. To the best of our knowledge, there is only one prospective study in Chinese patients using machine learning and metabolomics to predict T2DM incidence. No studies have been conducted on Caucasian patients or specifically addressed cardiovascular disease (CVD) patients. We thus believe that this is the first study in CVD patients.

  1. Another key statement made in paper is that the SVM algorithm is robust in diabetes prediction. To further consolidate this conclusion, I would suggest using the SVM model on previously published similar datasets to demonstrate its robuauthor-coverletter-36150382.v1.docxstness.

We agree with the reviewer, that it would be of great value to have further cohorts/datasets with similar variables. However, as already discussed in the original manuscript and just mentioned above, the metabolomic analysis is unique (targeted metabolomics with several kits) and quite expensive. We would be happy to have further data sets, but, unfortunately, we do not have such data with the same metabolomics done.

Reviewer 2 Report

Comments and Suggestions for Authors

Remarks to the Author:

The study conducts a thorough analysis using machine learning algorithms to predict the 4-year risk of developing type 2 diabetes mellitus (T2DM) by leveraging targeted quantitative metabolomics data. Initially comprising 279 cardiovascular risk patients without T2DM, the cohort underwent comprehensive baseline assessments, including anthropometric data and targeted metabolomics. Over the follow-up period, 11.5% of patients developed T2DM. Among the 362 variables measured, 77 features were selected using recursive feature elimination. The SVM model emerged as the top performer among conventional machine learning algorithms, exhibiting superior performance based on F1-score. However, there are major concerns regarding the comprehensiveness and novelty of this work, as outlined below.

Major comments:

1. This study didn’t seem to stand out. This paper is not the first one to use machine learning and metabolomics to predict the risk of T2DM, there are lot of similar papers that use metabolomics to predict T2DM. It’s not clear to me what’s the novelty of this paper. Authors need to highlight their unique contribution to the field.

In line 59, authors write “This study aims to evaluate ML's capacity to utilize metabolomic data for predicting T2DM onset.” If that’s the case, authors should consider conducting a meta-analysis that incorporates as many samples as possible. The sample characteristics available in this study seem inadequate for achieving this aim. For example, the study is based on a specific cohort of cardiovascular risk patients at very old age (median age 65), which may limit the generalizability of the findings to other populations, so perhaps the highlight should be the prediction of T2DM risk specifically among cardiovascular risk patients.

This point needs to be addressed significantly to clarify the novelty and unique contribution of the study to the existing literature on predicting T2DM using metabolomics and machine learning.

2. To me, it seems this study has a small sample size problem. In other metabolomics impact on diabetes studies like Lu et al. (2016) and Liu et al. (2017), the number of patients exceeds several thousand. Here, the total sample size is 279, with only 32 of them developing T2DM. After the training/testing split, only an average of 6.4 diabetes patients will appear in the testing set, which makes sensitivity and specificity estimation very inaccurate. It would greatly enhance the study if the authors could include more patients. Additionally, while the inclusion criteria specify only Caucasian patients with elevated cardiovascular risk undergoing coronary angiography, it would be beneficial to include patients from other racial backgrounds if available. This could allow for additional analyses to determine whether race impacts model performance.

3. Authors need to conduct further analysis on features. Regardless of whether they use Variable Importance (VarImp) or SHAP methods, which may not be easily understood by all readers, simpler methods should be employed to provide more intuitive information on feature importance. For instance, authors should perform association tests on each of the features individually, similar to what they did for baseline patient characteristics in Table 1. Another method to consider is lasso regression. The key point here is to provide an assessment of feature importance that has a clear meaning that most people can comprehend.

Furthermore, feature correlations should be calculated. At line 53, “ML algorithms excel at identifying subtle patterns and correlations within metabolomic data that traditional statistical 53 methods might miss”. However, there’s no such analysis performed given the author point out this important issue. Given that the number of features is larger than the number of samples, dimension reduction techniques are needed. While certain metabolomic features are identified as top predictors, the interpretation of these features and their biological significance may require further investigation.

4. Supplementary information was not correctly uploaded. Authors need to provide complete supplementary information for further assessment.

Minor comments:

1. Term “SHAP” was first mentioned in line 158, however complete definition was given at line 167 in figure caption and line 377. All abbreviations should be list full name as 1st time mentioned in main text.

2. In Figure 1, dots with different color should add a figure legend indicate what they represent.

3. In line 218, the authors conclude that anthropometric features like BMI play a less significant role compared to metabolomic data. However, I noticed that the range of BMI of patients included in this study has an interquartile range (IQR) from 25 to 31. According to Gray et al. (2015), when BMI is greater than or equal to 40, hazard ratios of T2DM show a significant increase. I doubt this conclusion is due to the lack of completeness of extreme values, thus the conclusion may not be comprehensive. Authors need to check their anthropometric distributions to see if their range is capable of detecting potential differences before making the conclusion that metabolomic data has superior power for T2DM prediction.

4. A narrowed selection of 77 variables is an important point, because it enables best model performance, it should be mentioned in abstract.

5. Since it’s a 4-year follow up studies, it would be interesting to learn if years of onsite can be also predicted.

6. Relevant to previous points, since diagnostics were made up to 4 years, will there be patients potentially develop T2DM in future? Will short follow up time impact model performance? Is longer follow up years needed?

7. At line 215, “Machine learning has now demonstrated for the first time that, alongside hexoses, these metabolite classes can predict the onset of diabetes in non-diabetic patients.” Is this the first paper that demonstrate machine learning can predict onset of diabetes? If its not please don’t make such over-statement.

8. Another key statement made in paper is that the SVM algorithm is robust in diabetes prediction. To further consolidate this conclusion, I would suggest using the SVM model on previously published similar datasets to demonstrate its robustness.

Author Response

(The authors gave the same response as above.)

Round 2

Reviewer 1 Report

Comments and Suggestions for Authors

The author has made a notable enhancement to the original manuscript. But, I think that the author has not adequately addressed the points raised in my comment. In particular, they have not cited the previous paper on a similar topic published by the corresponding author. The author indicated that the document in my comment was a conference abstract. However, upon further examination, it was determined to be an article with a DOI. It is recommended that this be further investigated to prevent the potential risk of plagiarism. Furthermore, they have not made the code available, which prevents the reader from reproducing their results. 

Author Response

Dear Editor, dear Reviewers,

Once again, we want to thank you for your very helpful comments. We are very grateful for all the suggestions made by both reviewers. Below, you will find our point-by-point response to your comments.

We now have included the reference mentioned by Reviewer#1.  We also added the R code used for analysis in the supplementary data. Furthermore, we have updated the figure and added a collinearity analysis to address the suggestions made by Reviewer#2.

In light of these additional revisions, we hope the manuscript is suitable for publication in its present form.

Sincerely,

Andreas Leiherer

Editor

[no further suggestions]

Reviewer#1

The author has made a notable enhancement to the original manuscript. But, I think that the author has not adequately addressed the points raised in my comment. In particular, they have not cited the previous paper on a similar topic published by the corresponding author. The author indicated that the document in my comment was a conference abstract. However, upon further examination, it was determined to be an article with a DOI. It is recommended that this be further investigated to prevent the potential risk of plagiarism. Furthermore, they have not made the code available, which prevents the reader from reproducing their results.

Thank you for your comment. Sorry for the confusion. We are now aware of the reference, the reviewer suggested we cite. This is an abstract, which I presented just a few weeks before at the ACC24. It is not a full paper yet but of course, an abstract published in JACC. In this project, we analyzed 894 consecutive patients referred to angiography. We used ceramide data we got from the Finnish group/company (Zora Biosciences) led by Rajo Laaksonnen. They measured a set of only 4 ceramides and developed a score called CERT which consists of 3 single molecules and 3 ratios (Hilvo et al. “Development and validation of a ceramide- and phospholipid-based cardiovascular risk estimation score for coronary artery disease patients”EurHeartJ (2020) 41 371-380 (supplements)). This score has proven effective predicting the cardiovascular risk - also with our patients. We were thus interested in determining whether this score, or specific components of it, though originally developed for assessing cardiovascular risk, could also be useful for predicting T2DM.

Given that there are hundreds of ceramides (which can potentially be measured if the methods are both functional and affordable), it seems likely to us that some of them may be linked to diabetes. Unfortunately, Rajo Laaksonnen measures only a preselected set for CERT comprising 3 ratios. Nevertheless, we found an association with one ratio (and we wrote the mentioned abstract for the ACC24) and we realized, that (some) ceramides may indeed be valuable in T2DM prediction. This result was in part also a basis for investigating more ceramides and more metabolites in general which we realized with the metabolomics assay (Biocrates, Austria) and the present paper. You see, there is no concern about plagiarism. Moreover, these were different patients and different core units doing the testing (Zora Biosciences vs. Biocrates). We feel that the ACC24 abstract has its special value but we did not want to oversize these findings because there were also other non-ceramide metabolites identified as important features. Nevertheless, in the revised version of the manuscript, we now cite this reference (“ref 18”), as suggested. In addition, we also modified the respective paragraph (discussion, lines 272-278) referring to the metabolomic features, which have been identified or corroborated in our study, mentioning that further investigation is required.

Regarding the code, in the revised manuscript, we provide the code used for our analyses of Multicollinearity, Lasso Regression, and ML including SHAP. 

Reviewer#2

Most of my comments has been addressed. I only have 2 more suggestions

  1. For figure 1, as I previously suggested, dots with different color should add a legend (within the figure, not text outside figure) indicate what they represent. I know author has the information in figure legend text, but for better presentation purpose, a good figure should able to explain itself as much as possible. If you can convey all information on figure itself without help of additional text you should do it.

The reviewer is right. We modified the figure accordingly, and now, it is much easier to understand. Thank you for your suggestion.

  1. Regarding analysis on features, “feature correlations should be calculated” I meant correlation between features not correlation between features and outcome. What I’m interested to see is wither interactions between features contributing to prediction power. The current feature analysis still focusing on individual feature importance. subtle patterns and correlations within metabolomic data should be identified.

Thank you again for your comment. Interactions of features contribute to the predictive power of an ML model, particularly the more complex ones that inherently account for interactions as this is the case for SVM due to the way they are constructed (e.g. Kernel Trick). The feature importance scores from such ML models tell not just about the individual importance but also hint at possible interactions if features are often used together in decision paths. SHAP analysis e.g. tries to account for this issue, that’s why we integrated it. I just read an interesting blog post by Bruce Desmarais  (“The first step in understanding the interpretability of ML methods is to recognize that a model needn’t distill the data down to simple linear relationships to provide clear substantive insights regarding the underlying process. On the contrary, while simple models may be easily interpretable, those interpretations can easily be wrong or misleading.”). Nevertheless, due to the “black box situation” in complex ML models, they are difficult to interpret. It was a very good idea of reviewer #2 to compare our best (complex) model with the linear (easier) Lasso Regression model, which does not use feature interactions. Therefore, in the second revision, we now additionally calculate the multicollinearity in terms of VIF for each feature. Multicollinearity, however, can be problematic for some types of models in particular linear regression models. LASSO-Regression does feature selection by shrinking coefficients of collinear features to zero. Therefore, it skips all features with high collinearity and just keeps those with low collinearity.

On the other hand, high collinearity is less of an issue for complex ML models (SVM). The benefit of keeping those features and taking into account interaction of (those) features, might result in the better performance of complex models. In the revised manuscript, we have attempted to present all this aforementioned information concisely and succinctly, but without overwhelming the reader and without making over-statements concerning our data (results, lines 149-151 and 203-205; discussion 249-251, 282-287;  supplementary table 2).

Reviewer 2 Report

Comments and Suggestions for Authors

Most of my comments has been addressed. I only have 2 more suggestions

1.     For figure 1, as I previously suggested, dots with different color should add a legend (within the figure, not text outside figure) indicate what they represent. I know author has the information in figure legend text, but for better presentation purpose, a good figure should able to explain itself as much as possible. If you can convey all information on figure itself without help of additional text you should do it.

2.      Regarding analysis on features, “feature correlations should be calculated” I meant correlation between features not correlation between features and outcome. What I’m interested to see is wither interactions between features contributing to prediction power. The current feature analysis still focusing on individual feature importance.  subtle patterns and correlations within metabolomic data should be identified.

Author Response

(The authors gave the same response as above.)

Round 3

Reviewer 1 Report

Comments and Suggestions for Authors

I believe the editorial office has addressed my concerns regarding this review.